

# BioShake: a Haskell EDSL for bioinformatics workflows

Justin Bedő

Bioinformatics Division, The Walter and Eliza Hall Institute, Parkville, VIC, Australia
Department of Computing and Information Systems, The University of Melbourne, Parkville, VIC, Australia

## ABSTRACT

Typical bioinformatics analyses comprise of long running computational workflows. An important part of reproducible research is the management and execution of these workflows to allow robust execution and to minimise errors. BioShake is an embedded domain specific language in Haskell for specifying and executing computational workflows for bioinformatics that significantly reduces the possibility of errors occurring. Unlike other workflow frameworks, BioShake raises many properties to the type level allowing the correctness of a workflow to be statically checked during compilation, catching errors before any lengthy execution process. BioShake builds on the Shake build tool to provide robust dependency tracking, parallel execution, reporting, and resumption capabilities. Finally, BioShake abstracts execution so that jobs can either be executed directly or submitted to a cluster. BioShake is available at http://github.com/PapenfussLab/bioshake.

## BACKGROUND

Bioinformatics workflows are typically composed of numerous programs and stages coupled together loosely by intermediate files. These workflows tend to be quite complex and require much computational time, hence a good workflow must be able to manage intermediate files, guarantee rentrability—the ability to re-enter a partially run workflow and continue from the latest point—and also facilitate easy description of workflows.

We present *BioShake*: a Haskell Embedded Domain Speciic Language (DSL) (EDSL) for bioinformatics workflows. The use of a language with strong types gives our framework several advantages over existing frameworks (*Amstutz et al., 2016*; *Di Tommaso et al., 2017*; *Goodstadt, 2010*; *Köster & Rahmann, 2018*; *Leipzig, 2016*; *OpenWDL, 2012*; *Vivian et al., 2017*) (see Table 1 for a high-level comparison):

1. The type system is strongly leveraged to prevent errors in the workflow construction *during compilation*—that is during development of the workflow (compile time) and prior to actual execution of the workflow (execution time). Workflows that pass the type checking at compile time are guaranteed free of certain classes of errors that with other workflow frameworks would ordinarily only be detected during execution. Catching errors significantly earlier in the analysis process reduces debugging time, which is especially advantages for bioinformatics as the workflows tend to have long runtimes.

Corresponding author
Justin Bedő, cu@cua0.org

Table 1 **High level feature comparison of BioShake with other execution engines (Toil, Cromwell), specification languages (WDL, CWL), and DSLs (NextFlow, Snakemake).** Dashes indicate that feature is not applicable.

| | Snakemake | NextFlow | Toil | Cromwell | WDL | CWL | BioShake |
|---|---|---|---|---|---|---|---|
| DSL | ✓ | ✓ | – | – | ✓ | ✓ | |
| Embedded DSL | | | – | – | | | ✓ |
| Python | | | ✓ | – | | | |
| Strong static typing | | | | – | | | ✓ |
| Type inferencing | | | | – | | | ✓ |
| Extrinsic specification | ✓ | | | – | | | ✓ |
| Intrinsic specification | | ✓ | ✓ | – | ✓ | ✓ | ✓ |
| Functional language | | | | – | | | ✓ |
| Container integration | | | ✓ | ✓ | – | – | |
| Cloud computing integration | ✓ | ✓ | ✓ | ✓ | – | – | |
| Cluster integration (Torque) | ✓ | ✓ | ✓ | ✓ | – | – | ✓ |
| Cluster integration (Slurm) | ✓ | ✓ | ✓ | ✓ | – | – | |
| Cluster integration (SGE) | ✓ | ✓ | ✓ | ✓ | – | – | |
| Cluster integration (LSF) | ✓ | ✓ | ✓ | | – | – | |
| Cluster integration DRMAA | ✓ | | | | – | – | |
| Direct execution | ✓ | ✓ | ✓ | ✓ | – | – | ✓ |

To the best of our knowledge, this is the first bioinformatics workflow framework to use strong typing and type inference to prevent specification errors during compile time.

2. Naming of outputs at various stages of a workflow are abstracted by BioShake. Output at a stage can be explicitly named if they are desired outputs. Thus, the burden of constructing names for temporary files is alleviated. This is similar in spirit to *Sadedin, Pope & Oshlack (2012)* who also allow abstraction away from explicit filenames.

3. BioShake builds on top of *Shake*, an industrial strength build tool also implemented as an EDSL in Haskell. BioShake thus inherits the reporting features, robust dependency tracking, and resumption capabilities offered by the underlying Shake architecture.

4. Unlike underlying Shake that expects *dependencies* to be specified (i.e., in a DAG the arrows point from the target back towards the source(s)), BioShake allows forward specification of workflows (i.e., the arrows point forward). As bioinformatics workflows tend to be quite long and mostly linear, this eases the cognitive burden during workflow design and improves readability.

5. Non-linear workflows are constructed using typical Haskell constructs such as maps and folds. Combinators are available for the most common grouping of outputs together for a subsequent stage. However, as the main data type is recursively defined, outputs of a stage can always be referenced by subsequent stages without explicit non-linear constructs (i.e., the alignments used for variant calling are available for a subsequent variant annotation stage without explicitly introducing non-linearity).

BioShake, in essence, is an EDSL for specifying workflows that compiles down to an execution engine (Shake). In this respect, it resembles other specification languages such

as Common Workflow Language (CWL) (*Amstutz et al., 2016*) and Workflow Description Language (WDL) (*OpenWDL, 2012*), but executes on top of Shake. Table 1 provides a high level feature overview of BioShake when compared to several other workflow specification language, workflow EDSLs, and execution engines. We will further elaborate on the unique features of BioShake:

**Strong type-checking**   The use of a language with strong types gives our framework several advantages over existing frameworks (*Amstutz et al., 2016*; *Di Tommaso et al., 2017*; *Goodstadt, 2010*; *Köster & Rahmann, 2018*; *Leipzig, 2016*; *OpenWDL, 2012*; *Sadedin, Pope & Oshlack, 2012*; *Vivian et al., 2017*). Our framework leverages Haskell's strong type-checker to prevent many errors that can arise in the specification of a workflow. As an example, file formats are statically checked by the type system to prevent specification of workflows with incompatible intermediate file formats. Furthermore, tags are implemented through Haskell type-classes to allow metadata tagging, allowing various properties of files—such as whether a Binary Alignment Map (BAM) file is sorted—to be statically checked. Thus, a misspecified workflow will simply fail to compile, catching these bugs well before the lengthy execution. As tags are represented in the type system, they are assumed to hold and are not validated at runtime (i.e., an output tagged as sorted and in Binary Alignment Map (BAM) format is not verified to be sorted nor a valid Binary Alignment Map (BAM) file at runtime). Consequently, no runtime overhead is incurred.

**Intrinsic and extrinsic building**   Our framework builds upon the Shake EDSL (*Mitchell, 2012*), which is a make-like build tool. Similarly to make, dependencies in Shake are specified in an *extrinsic* manner (called implicit by *Leipzig, 2016*); that is, a build rule will define its input dependencies based on the output file path. Our EDSL compiles down to Shake rules, but allows the specification of workflows in an *intrinsically*, whereby the processing chain is explicitly stated and hence no filename based dependency graph needs to be specified. However, as BioShake compiles to Shake, both extrinsic and intrinsic rules can be mixed, allowing a choice to be made to maximise workflow specification clarity. For example, small "side" processing like generation of indices can be specified extrinsically, removing the need for an explicit index step in the workflow specification.

Furthermore, the use of explicit sequencing for defining workflows allows abstraction away from the filename level: intermediate files can be automatically named and managed by BioShake, removing the burden of naming the intermediate files, with only desired outputs requiring explicit naming.

The following examples illustrate various aspects of the BioShake EDSL. Examples 1 and 2 demonstrate the expression of a couple of workflows in the EDSL. Example 3 shows how an end-user interface may be constructed for end-users rather than workflow designers.

---

**Example 1.**   (**Simple sequencing workflow**)

*The following is an example of a workflow expressed in the BioShake EDSL:*
*align → fixMates → sort → markDups → call → out* [`"output.vcf"`]
*From this example it is clear what the stages are, and the names of the files flowing between stages is implicit and managed by BioShake. The exception is the explicitly named*

---

*output, which is the output of the whole workflow. Non-linearity is handled by constructors that accept the extra inputs, but workflows can always refer backwards along → to retrieve prior build products (e.g., to fetch Binary Alignment Map (BAM) files used to generate a set of variant calls), reducing the need for non-linearity. A full example of this workflow is given in Appendix A: Full code for Example 1.*

---

**Example 2.** (**Scatter–gather workflow**)

*BioShake supports scatter-gather type workflows and provides functions to join and split the output of stages. An example is provided in the BioShake repository with the following workflow:*
let
*chunks = split (query → chunk n)*
**in**
*withAll (map (λ c → c → blast → extract) chunks) concat → out* ["seqs.txt"]
*A FastA query file is sharded in* n *chunks, blast executed on each chunk concurrently with the top sequences extracted and concatenated into the final named output. The* split *and* with All *functions are provided by BioShake and splits the outputs of a stage into a list of individual stages, and withAll is the inverse function.*

---

**Example 3.** (**Providing workflows to end-users**)

*Examples 1 and 2 are examples of workflows from a designer's perspective not an end-user's perspective, however as BioShake is an EDSL it is ideally suited to providing end-users with end-to-end workflows. A command line, graphical, or web interface can be designed using other Haskell packages to allow users to input the location and structure of their data, with the builds then being produced by BioShake. The interface, workflow, and build system can then be compiled to a single binary for distribution to users. For example, a generalisation of the example 1 workflow to multiple samples could be compiled to an executable* call *with a command line interface*
call [-t nthreads] [-o output VCF] fq1 fq2 ...

---

## IMPLEMENTATION

### Core data types
BioShake is built using a tagless-final style (*Carette, Kiselyov & Shan, 2009*) around the following datatype:
> **data** $a \to b$
> **where**
> ($\to$):: $a \to b \to a \to b$
> **infixl** $1 \to$

This datatype represents the conjunction of two stages *a* and *b*. As we are compiling to Shake rules, the `Buildable` class represents a way to build thing of type *a* by producing Shake actions:

**class** *Buildable a*

**where**

*build:: a → Action ()*

Finally, as we are ultimately building files on disk, we use a typeclass to represent types that can be mapped to filenames:

**class** *Pathable a*

**where**

*paths:: a → [FilePath]*

### Defining stages

A stage—for example `aligning` and `sorting`—is a type in this representation. Such a type is an instance of `Pathable` as outputs from the stage are files, and also `Buildable` as the stage is associated with some Shake actions required to build the outputs. We give a simple example of declaring a stage that sorts bam files in Example 4.

---

**Example 4.** (**Stage definitions**)

*Consider the stage of sorting a bed file using samtools. We first define a datatype to represent the sorting stage and to carry all configuration options needed for sorting:*

```
data Sort = Sort
```

*This datatype must be an instance of* `Pathable` *to define the filenames output from the stage. Naming can take place according to several schemes, but here we will opt to use hashes to name output files. This ensures the filename is unique and relatively short.*

**instance** *Pathable a ⇒ Pathable (a → Sort)*

**where**

*paths (a → _) =* **let**

*inputs = paths a*

**in**

*[hash inputs ++* `".sort.bed"`*]*

*In the above, hash:: Binary a ⇒ a → String is a cryptographic hash function such as sha1 with base32 encoding. Many choices are appropriate here.*

*Finally, we describe how to sort files by making* `Sort` *an instance of* `Buildable`:

**instance** *(Pathable a, IsBam a) ⇒ Buildable (a → Sort)*

**where**

*build p@(a → _) =* **let**

*[input] = paths a*

*[out] = paths p*

**in**

*cmd* `"samtools sort"` *[input] [*`"-o"`*, out]*

---

> *Note here that* `IsBam` *is a precondition for the instance: the sort stage is only applicable to Binary Alignment Map (BAM) files. Likewise, the output of the sort is also a Binary Alignment Map (BAM) file, so we declare that too:*
> **instance** *IsBam (a → Sort)*
> *The tag* `IsBam` *itself can be declared as the empty typeclass class IsBam a. See section 2.5 for a discussion of tags and their utility.*

### Compiling to Shake rules

The workflows as specified by the core data types are compiled to Shake rules, with Shake executing the build process. The distinction between `Buildable` and `Compilable` types are that the former generate Shake `Actions` and the latter Shake `Rules`. The `Compiler` therefore extends the `Rules` monad, augmenting it with some additional state:

    **type** *Compiler = State T (S.Set [FilePath]) Rules*

The state here captures rules we have already compiled. As the same stages may be applied in several concurrent workflows (i.e., the same preprocessing may be applied but different subsequent processing defined) the set of rules already compiled must be maintained. When compiling a rule, the state is checked to ensure the rule is new, and skipped otherwise. The rule compiler evaluates the state transformer, initialising the state to the empty set:

    *compileRules :: Compiler () → Rules ()*
    *compileRules p = evalStateT p mempty*

A compilable typeclass abstracts over types that can be compiled:

    **class** *Compilable a*
    **where**
    *compile:: a → Compiler ()*

$a → b$ is `Compilable` if the input and output paths are defined, the subsequent stage $a$ is `Compilable`, and $a → b$ is `Buildable`. Compilation in this case defines a rule to build the output paths with established dependencies on the input paths using the `build` function. For convenience BioShake provides a `compileAndWant` function that both compiles a workflow to *Rules* and requests Shake to build it.

    These rules can then be executed by Shake using the (simplified type for clarity) function
    *bioShake :: Rules () → IO ()*

### Dynamic workflows

Dynamic workflows are those where the number or type of output files for a stage is dependent on the input to the stage. The BioShake abstractions do not directly allow this, however it is still possible by splitting the workflow into two static workflows with Haskell logic in-between. Example 5 demonstates this approach.

---

> **Example 5.** (**Dynamic workflows**)
>
> *Suppose a workflow assembled contigs from FastQ files and then processed each contig separately, i.e., the workflow is dynamic and dependent on the number of contigs assembled. This could be expressed as two static builds with the dynamic logic handled in Haskell:*
>
> **let**
> *make = bioShake ∘ compileAndWant*
> **in**
> *make (inputs → assemble → out* [`"assembly.fa"`]*)*
> *nContigs ← fmap (length ∘ tail ∘ splitOn* `">"`*) (readFile* `"assembly.fa"`*)*
> *make (map (λ c → c → processContig) (shard* `"assembly.fa"` *nContigs))*
> *After* `assembly.fa` *is produced by the first workflow the number of contigs is fixed and the second workflow can shard the contigs into individual files and process them independently.*

## Tags

BioShake uses tags to ensure type errors will be raised if stages are incompatible. Other workflow systems such as CWL have limited ability to tag outputs and verify inputs using file patterns, however this approach does not scale gracefully to many tags and tags carrying additional metadata. As encoding tags into filenames inherently imposes an ordering between tags, parsing and generation of names is complicated, and with many tags filesystem limitations on length can be encountered.

We have already seen in Example 4 the use of *IsBam* to ensure the input file format of *Sort* is compatible. By convention, BioShake uses the file extension prefixed by *Is* as tags for filetype, e.g.,: *IsBam*, *IsSam*, *IsVCF*. Other types of metadata are used such as if a file is sorted (*Sorted*) or if duplicate reads have been removed (*DeDuped*) or marked (*DupsMarked*). These tags allow input requirements of sorting or deduplication to be captured when defining stages. It is easy to state implications, for example to consider files to have duplicates marked if duplicates have been removed:

**instance** *Deduped a ⇒ DupsMarked a*

Properties, where appropriate, can also automatically propagate down the workflow; for example, once a file is *DeDuped* all subsequent outputs carry the *DeDuped* tag: **instance** *Deduped a ⇒ Deduped (a → b)*

Finally, the tags discussed so far have been empty type classes, however tags can easily carry more information. For example, BioShake uses a *Referenced* tag to represent the association of a reference genome. This tag is defined

**class** *Referenced*
**where**
*getRef:: FilePath*
**instance** *Referenced a ⇒ Referenced (a → b)*

and allows stages to extract the path to the reference genome, automatically propagating down the workflow allowing identification of the reference at any stage.
## EDAM ontology

EDAM (*Ison et al., 2013*) is an ontology containing terms and concepts that are prevalent in bioinformatics. As it is a formal ontology, the terms are organised into a hierarchical tree structure, with each term containing reference to parent terms. EDAM can be used with the flat tagging structure introduced in the previous section using Template Haskell (TH)—a metaprogramming language allowing Haskell code to be generated during compilation—to establish the tree. This is different from other systems that support EDAM for symantic annotation such as CWL as the ontology is represented at the type level and hence prevents a class of errors.

BioShake provides the EDAM ontology in the EDAM module. This module provides EDAM terms identified by their short name, along with some Template Haskell (TH) for associating EDAM terms to types. For example, the *FASTQ-illumina* term (http://edamontology.org/format_1931) is represented by the tag *FastqIllumina* and a type can be tagged using the is template Haskell function, for example:

> **import** *BioShake.EDAM*
> **data** *MyType = MyType*
> $(*is* ''`MyType` ''`FastqIllumina`)

The *is* function declares the given type to be instances of all parents of the EDAM term, allowing tag matching at any level in the hierarchy. For instance, `MyType` in the above example will also carry the `Fastq` tag as `FastqIllumina` is a child of `Fastq`. EDAM types can be used similarly to tags as described in section 'Tags'.

## Abstracting the execution platform

In Example 4, the Shake function *cmd* is directly used to execute samtools and perform the build, however it is useful to abstract away from *cmd* directly to allow the command to be executed instead on (say) a cluster, cloud service, or remote machine. BioShake achieves this flexibility by using free monad transformers to provide a function *run*—the equivalent of *cmd* –but where the actual execution may take place via submitting a script to a cluster queue, for example.

To this end, the datatype for stages in BioShake are augmented by a free parameter to carry implementation specific default configuration –e.g., cluster job submission resources. In the running example of sorting a bed file, the augmented datatype is data `Sort c =` `Sort c`.

## Reducing boilerplate

Much of the code necessary for defining a new stage can be automatically written using Template Haskell (TH) leading to succinct definitions of stages, increasing clarity of code and reducing boilerplate. BioShake has Template Haskell (TH) functions for generating instances of *Pathable* and *Buildable*, and for managing the tags. Example 6 shows how Template Haskell (TH) is used to simplify the stage definition presented in Example 4.

---

**Example 6.** (**Dynamic workflows**)

*Template Haskell (TH) can simplify Example 4 considerably. First we have the augmented type definitions:*

**data** *Sort c = Sort c*

*The instances for Pathable and the various tags can be generated with the Template Haskell (TH) splice*

$(*makeTypes* ''`Sort` [''`IsBam`, ''`Sorted`] [ ])

*This splice generates a Pathable instance using the hashed path names, and also declares the output to be instances of IsBam and Sorted. The first tag in the list of output tags determines the file extension. The second empty list allows the definition of transient tags; that is the tags that if present on the input paths will hold for the output files after the stage. Finally, given a generic definition of the build*

*buildSort nThreads _ (paths → [input]) [out] =*

*run* "`samtools sort`" [*input*] ["`-@`", *show nThreads*] ["`-o`", *out*]

*the Buildable instances can be generated with the splice*

$(*makeThreaded* ''`Sort` [''`IsBam`] '`buildSortBam`)

*This splice takes the type, a list of required tags for the input, and the build function. Here, the build function is passed the number of threads to use, the Sort object, the input object and a list of output paths.*

## RESULTS AND DISCUSSION

We have presented a framework for describing and executing bioinformatics workflows. The framework is an EDSL in Haskell built on Shake, allowing us to leverage the robustness of Shake and also the power of Haskell's type system to prevent many types of errors in workflow construction. Preventing errors before execution is particularly beneficial for bioinformatics workflows as they tend to be long running and thus catching errors during compile reduces the debugging time significantly.

Though this library is built around Shake as the execution engine, the core value lies in the unique abstraction and use of types to capture metadata. It is feasible to compile a specification to a different backend instead of Shake, such as Toil (*Vivian et al., 2017*) or Cromwell (*Cromwell, 2015*) via CWL (*Amstutz et al., 2016*) or WDL (*OpenWDL, 2012*). This would allow leveraging of the cloud and containerisation facilities of Toil and Cromwell. The abstraction used may also be useful in other domains where long data-transformation stages are applied, such as data mining on large datasets.

Though many errors are currently caught by the type system, there are still classes of errors that are not. Notably, the Pathable class instance maps stages to lists of files with unknown length. Thus, the number of files expected to be exchanged between two stages may differ, causing a runtime error. Lists of typed length could be used to prevent this error, however this would increase the complexity for users. BioShake attempts to strike a balance between usability and type safe guarantees.

## CONCLUSIONS

We have presented a unique EDSL in Haskell for specifying bioinformatics workflows. The Haskell type checker is used extensively to prevent specification errors, allowing many errors to be caught during compilation rather than runtime. To our knowledge, this is the first bioinformatics workflow framework in Haskell, as well as the first formalisation of bioinformatics workflows and their attributes in a type system from the Hindley–Milner family.

## A: FULL CODE FOR EXAMPLE 1

This is a simple pipeline to demonstrate how to specify pipelines using the bioshake framework. It will accept pairs of fastq files from the command line, align them against a reference sequence with BWA, then call variants on all asamples using the platypus variant caller.

```
> import Bioshake
> import Control.Monad
> import Data.List.Split
> import Development.Shake
> import Development.Shake.FilePath
> import System.Environment
> import System.Exit
```

We will align reads using BWA, sort and filter with samtools, and finally call with Platypus

```
> import Bioshake.BWA as B
> import Bioshake.Platypus
> import Bioshake.Samtools as S
```

First, define a datatype to represent our paired-end samples on disk.

```
> data Sample = Sample FilePath FilePath
> instance Show Sample where
> show (Sample a b) = takeFileName (dropExtensions a) ++ ”-”
> ++ takeFileName (dropExtensions b)
```

The default instance of Compilabe suffices for files that already exist on disk and that do not require building

```
> instance Compilable Sample
```

Bioshake uses type classes to encode properties about stages in the pipeline. The first property we're going to declare is that these samples are paired end reads. We also will declare that the input files are FastQ files.

```
> instance PairedEnd Sample
> instance IsFastQ Sample
```

Next, we need to declare which reference the type is going to be associated with. This involves instantiating the Referenced class and declaring the path to the reference and the short name of the reference:

```
> instance Referenced Sample where
> getRef _ = "ref.fa"
> name _ = "SL1344"
```

Finally, we describe how to get the paths for a Sample, which in this case is extracted from our Sample datatype:

```
> instance Pathable Sample where
> paths (Sample a b) = [a, b]
```

Command line arguments are navly parsed: we simply assume each pair of arguments are paths to the two paired-end FastQ files.

```
> parseArgs = map (/[a, b] -> Sample a b) . chunksOf 2
> main = do
> args <- getArgs
> when (null args || length args 'mod' 2 /= 0)$ do
> putStrLn "error: expecting paired fastq files as input"
> exitFailure
> let samples = parseArgs args
```

The number of threads used has to be specified to bioshake in two ways: the number of threads used for each stage, and the maximum number of concurrent threads in total. Threads per job can be specified by giving a Threads instance to each stage, or at a higher level. Here we give Threads 1, meaning stages run single threaded, and limit the maximum number of concurrent threads to 2 in total.

```
> withArgs [] $ give (Threads 1)$ bioshake 2 shakeOptions $ do
```

bioshake, like shakeArgs, expects Shake Rules. We can therefore want thing and define standard Shake Rules as normal. In this case we want our output vcf file, which we'll call "calls.vcf".

```
> want ["calls.vcf"]
```

In addition to that, we will bring into scope the rules for indexing bamfiles (building .bam.bai from .bam) using samtools, and similarly for the BWA indexing rules.

```
> B.indexRules
> S.indexRules
```

Finally, we compile our workflow down to Shake Rules.

```
> compileRules$
```

We have one simple workflow in this case. Alignment and processing is first applied to each individual sample. The samples are then pooled and called as a group using Platypus.

```
> let aligned = map (/s -> s :-> align
> :-> fixMates
> :-> sortBam
> :-> markDups
> :-> addRGLine (show s)) samples in
> compile$ withAll aligned :-> call :-> out ["calls.vcf"]
```

## ACKNOWLEDGEMENTS

I thank Tony Papenfuss for supporting this work and helpful discussions. I also thank Leon di Stefano and Jan Schröder for helpful discussions.

### Funding

Justin Bedő is supported by a WEHI Centenary Fellowship in Rare Cancer funded by the Stafford Fox Medical Research Foundation. The funders had no role in study design, data collection and analysis, decision to publish, or preparation of the manuscript.

### Grant Disclosures

The following grant information was disclosed by the author:
Stafford Fox Medical Research Foundation.

### Competing Interests

The authors declare there are no competing interests.

### Author Contributions

- Justin Bedő conceived and designed the experiments, performed the experiments, analyzed the data, contributed reagents/materials/analysis tools, prepared figures and/or tables, authored or reviewed drafts of the paper, approved the final draft.

### Data Availability

Data is available at GitHub: http://github.com/PapenfussLab/bioshake.

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
