# Peer review of "BioShake: a Haskell EDSL for bioinformatics workflows"

_PeerJ, doi:10.7717/peerj.7223_

## Round 0.1 · original submission · Major Revisions

Dear Dr. Bedő:

Thanks for submitting your manuscript to PeerJ. I have now received three independent reviews of your work, and as you will see, the reviewers raised some concerns about the research. Despite this, these reviewers are optimistic about your work and the potential impact it will lend to research on computational biology and bioinformatics. Thus, I encourage you to revise your manuscript accordingly, taking into account all of the concerns raised by the reviewers. Please remember to use the marked-up manuscript kindly provided by reviewer 1.

While the concerns of the reviewers are relatively minor, this is a major revision to ensure that the original reviewers have a chance to evaluate your responses to their concerns.

In your revision, please ensure that Bioshake usability is not an issue (is there another approach aside from Nix?). I would like for the reviewers to be able to successfully implement Bioshake prior to publication of your work. Also, consider adding more than one empirical dataset/example workflow.

I look forward to seeing your revision, and thanks again for submitting your work to PeerJ.

Good luck with your revision,

-joe

·

Basic reporting

no comment

Experimental design

no comment

Validity of the findings

no comment

Additional comments

Attached annotated PDF contains general comments and addresses grammatical issues.

I really like the idea of leveraging a strong type system to validate workflows during construction, nicely done!

I didn't see it mentioned anywhere, but does Bioshake support any sort of dynamic workflow structure? E.g. if you wanted to run one tool per chromosome, but the number of chromosomes changes depending on the input, can those jobs be generated dynamically or does the entire graph have to be compiled statically prior to execution?

You mention, "It is feasible to compile a specification to a different backend instead of Shake, such as Toil [...]." Do you have plans to do this or is this an area of active work? This would get around the issue of Bioshake lacking any type of cloud support.

I think the text needs to be tightened up a bit before submission. Both to address small grammatical issues as well as to avoid repetition between sections. I like the bullet points in the background for succinctly describing the advantages of your tool, but that information is essentially repeated later on without much more additional information presented to the reader.

·

Basic reporting

Please make the following changes:

Line 14:"Bioshake is an embedded domain specific language embedded in Haskell" -> "Bioshake is a domain specific language embedded in Haskell"

Line 78: Dependencies in shake are specified in an extrinsic manner "called internal/external" -> "called implicit"

Experimental design

I also request you make the following changes/enhancements:
- You need to briefly define the term "template Haskell". Not many readers outside the Haskell community would know what that means.
- You should address the most popular DSL frameworks in your Table 1, which according to this recent poll by Albert Vilella https://docs.google.com/spreadsheets/d/1plkAsT_S3CzSeb7ivxyjRnHyrK3JclUCXeUMf_azraY/ are NextFlow and Snakemake by a huge margin. Ruffus is now seldom used and should be removed.
- The bioshake github repository uses git-lfs to store the sequence files. If you try to simply clone the repo without having git-lfs installed, the pipeline breaks mysteriously. 99.999% of git users do not have git-lfs installed. Once that is fixed, the example runs without errors but Platypus produces a VCF that has a header but no entries. Please warn the user they will need git-lfs and come up with some example files that actually produce a variant call.

Validity of the findings

Bioshake does implement some significant advances over the state-of-the-art. Bioshake introduces a strong typing mechanism, inference, and support for the EDAM ontology. It is also the first Haskell-based DSL. Haskell is a fast and perhaps underrated functional language in the bioinformatics community. The advanced type system offered by Haskell is leveraged in Bioshake to implement compile-time validation of workflows, which is novel for frameworks.

While these are impressive accomplishments, I don't want the manuscript to oversell these features so much that they confuse the reader. Please add the following disclaimers:
- Put a disclaimer that configuration-based workflow languages CWL, do have some primitive mechanisms for matching inputs and outputs between tasks, although these are accomplished using file suffix patterns and not a formal language type convention as implemented in Bioshake. You can then explain how your system allows for hierarchy and faceting (such as "sorted").
- Put a brief disclaimer that Bioshake does not actually check the file output contents for conformance to these types. It doesn't actually parse SAM files to make sure they are sorted. (This might seem obvious to you but I think you might give this impression to some readers.)
- Mention that CWL support EDAM entities in tool definitions, but this is mostly for semantic annotation, not type-checking

·

Basic reporting

The author should be consistent with the capitalisation of "Shake" throughout the manuscript.

The difference between "compile time" and "execution time" may not be obvious to some of the target audience. I suggest it is made clear early in the paper that "compile time" refers to the development of the pipeline and that a pipeline that passes these checks will be guaranteed free of many classes of errors that only appear the user runtime of other workflow frameworks.

The formatting of the titles for "Example 1" and "Example 2" are difficult to find on scanning the manuscript, I suggest these be in bold.

line 83: s/make/made/

line 94: replace "recurse" with "refer". I think "recurse" necessarily implies self-referential which I don't think is the case here.

line 105: s/build/built/

line 119: s/ensure/ensures/

example after line 195, I suggest the variable "t" be replaced with a more meaning variable such as "nthreads"

line 141: I am not clear what is meant here by "only compiled if they do not already exist". Is this referring to during the construction of workflow?

The paper includes basic examples from the point of view of a developer of a new workflow. There should also be an example of how such a workflow would be used by a end-user, particularly with multiple input files and requested threads.

The code near line 141 is difficult to understand - for example including an operator "&%>" that is not defined anywhere. I'd suggest removing this, or replacing with pseudo-code.

Experimental design

Table 1 has a highlevel comparison to other similar tools/frameworks. I recommend the author include frameworks that are more common to the intended bioinformatics audience, such as snakemake & nextflow.

I do not understand the difference between "$(is ...)" and "$(isP ...") around line 168. Could the author clarify in the text the differences

Validity of the findings

This paper introduces Bioshake which is a useful contribution to workflow construction that allows development of workflows with less errors.

Additional comments

I commend the author on a new an interesting approach to workflow development for bioinformatics.

I think the paper would benefit from a more in-depth example of a workflow. And it would be nice to have more than one example on the github site at : https://github.com/PapenfussLab/bioshake/tree/master/examples

I was unable to test bioshake on my laptop as the documented build instructions require Nix which I cannot install as it requires admin privileges. I also failed to build using stackage, and have raised a corresponding issue on the github site.

---

## Round 0.2 · accepted · Accept

Dear Dr. Bedő:

Thanks for re-submitting your manuscript to PeerJ, and for addressing the concerns raised by the reviewers. I now believe that your manuscript is suitable for publication. Congratulations! I look forward to seeing this work in print, and I anticipate it being an important resource on computational biology and bioinformatics. Thanks again for choosing PeerJ to publish such important work.

-joe

·

Basic reporting

The author's revisions addressed all of my concerns regarding comparable tools and standards. The author also made substantial improvements to the inline examples in the manuscript and source code which I feel adequately address the comments from the other reviewers.

Experimental design

This is a well-designed and documented tool.

Validity of the findings

no comment

Additional comments

Great contribution!